# Consumption of Vitamin-A-Rich Foods and Vitamin A Supplementation for Children under Two Years Old in 51 Low- and Middle-Income Countries

**DOI:** 10.3390/nu14010188

**Published:** 2021-12-31

**Authors:** Omar Karlsson, Rockli Kim, Andreas Hasman, S. V. Subramanian

**Affiliations:** 1Takemi Program in International Health, Harvard T.H. Chan School of Public Health, Harvard University, Boston, MA 02115, USA; okarlsson@hsph.harvard.edu; 2Department of Economic History, School of Economics and Management, Lund University, 220 07 Lund, Sweden; 3Division of Health Policy & Management, College of Health Science, Korea University, Seongbuk-gu, Seoul 02841, Korea; 4Harvard Center for Population and Development Studies, Harvard University, Cambridge, MA 02138, USA; 5UNICEF Programme Division, New York, NY 10017, USA; ahasman@unicef.org; 6Department of Social and Behavioral Sciences, Harvard T.H. Chan School of Public Health, Harvard University, Boston, MA 02115, USA

**Keywords:** vitamin A supplements, dietary diversity, vitamin-A-rich foods, child nutrition, low- and middle-income countries

## Abstract

Vitamin A supplementation for children 6–59 months old is an important intervention that boosts immune function, especially where children do not consume enough vitamin-A-rich foods. However, the low coverage of vitamin A supplementation is a persistent problem in low- and middle-income countries. We first estimated the percentage of children 6–23 months old receiving the minimum dietary diversity, vitamin-A-rich foods, and vitamin A supplementation, and second, the difference in the percentage receiving vitamin A supplementation between children 6–23 months old and children 24–59 months old using nationally representative cross-sectional household surveys, namely, the Demographic and Health Surveys, conducted from 2010 to 2019 in 51 low- and middle-income countries. Overall, 22% (95% CI: 22, 23) of children received the minimum dietary diversity, 55% (95% CI: 54, 55) received vitamin-A-rich foods, 59% (95% CI: 58, 59) received vitamin A supplementation, and 78% (95% CI: 78, 79) received either vitamin-A-rich foods or supplementation. A wide variation across countries was observed; for example, the percentage of children that received either vitamin-A-rich foods or supplementation ranged from 53% (95% CI: 49, 57) in Guinea to 96% (95% CI: 95, 97) in Burundi. The coverage of vitamin A supplementation should be improved, especially for children 6–23 months old, in most countries, particularly where the consumption of vitamin-A-rich foods is inadequate.

## 1. Introduction

The 1000 days from conception until age two are critical for human development. Undernutrition is particularly detrimental before age two, with negative consequences for survival and cognitive development, as well as health, education, and income in the long term [1,2]. The supplementary feeding period starts at around six months old, when breastfeeding alone no longer provides enough nutrition; then, children should receive safe and nutritionally adequate food—in terms of protein and energy, as well as vitamins and minerals—together with continued breastfeeding until age 23 months or longer [3]. Globally, about 28% of children in low- and middle-income countries do not receive the minimum dietary diversity during the complementary feeding period [4]; and at ages 6–23 months, during the weaning period, only 19% of children are fed minimum acceptable diets. Consumption of vitamin-A-rich foods is insufficient in many places, and around 190 million children are vitamin A deficient according to most recent global estimates (however, from 2005 [5]). Vitamin A intake was found to be sufficient in most high-income countries, and therefore, the deficiency is presumably mostly in low- and middle-income countries [6]. Vitamin A is essential for growth and immune function, and deficiencies make children susceptible to infections and death due to depressed immune function [7,8,9].

Strategies to address vitamin A deficiency include supplementation, dietary diversification, and food fortification [10]. Utilizing the right mix of these strategies, depending on the national and subnational context and specific circumstances, nutrition programs help to improve child health outcomes [11,12,13]. For example, administering high-dose vitamin A supplements to children 6–59 months old every four to six months may reduce the risk of child mortality by 12–24% in communities suffering from vitamin A deficiencies [14,15]. However, interventions fail to reach full coverage, that is, young children 6–59 months old. In 2018, coverage with two annual doses of vitamin A supplementation was suboptimal (at 62% of children 6–59 months old) [16], and although more countries mandate food fortification, the minimum required amounts and compliance with regulation remain low in many countries [17]. Resource scarcity, ineffective program delivery, inadequate enforcement of regulations, as well as lack of awareness and capacity of governments and other actors to finance and implement nutrition interventions are blamed for the under-performance [18]. Disruptions from the COVID-19 pandemic have reduced coverage further [19,20].

Vitamin A supplementation is critical in communities where the consumption of vitamin-A-rich foods is deficient. In this analysis, we estimated the proportion of children 6–23 months old who received minimum dietary diversity, vitamin-A-rich foods, and vitamin A supplementation in 51 low- and middle-income countries surveyed in the Demographic and Health Surveys 2010–2019. Our primary outcome showed the proportion of children that were either fed vitamin-A-rich foods or received vitamin A supplementation; this was important since children that receive neither are at a particularly high risk of vitamin A deficiency and the associated adverse outcomes. We broke down these results by sex, household living standards, and urban–rural residency. In most countries, vitamin A supplementation is reported for and targeted at children 6–59 months old. In this study, we also estimated the percentage of children 24–59 months who received vitamin A supplementation and compared this with the coverage in children 6–23 months old; the early life period—during the first 1000 days—when the consequences of undernutrition are particularly harmful to survival and human development. Feeding data was not available for children 2–4 years old. We show results for each country, as well as aggregated by region and income groups.

## 2. Materials and Methods

### 2.1. Data

Our data came from the Demographic and Health Surveys, which are conducted regularly in numerous low- and middle-income countries. The inclusion criterion was all surveys recording information on feeding and vitamin A supplementation. We included the most recent survey for each country but excluded countries with no surveys conducted after 2010. Out of the 90 countries that have participated in the DHS since 1984, our selection criterion resulted in 51 surveys conducted from 2010 to 2019 (see Appendix A for countries and survey years and Appendix A for sample sizes).

The Demographic and Health Surveys use standardized questionnaires translated into local languages. A multi-stage stratified sampling design is used to obtain nationally representative data on the health of women and children [21]. Stratification is generally based on administrative or geographic subnational regions, which are further separated into urban and rural areas. Primary sampling units, which usually consist of villages or neighborhoods, are sampled from these strata with a probability that is proportional to their size. All households are listed from the selected primary sampling units, and then 20–30 households are selected using equal probability systematic sampling [22]. Several questionnaires are administered; for example, a questionnaire that records the characteristics of the household and a questionnaire for women 15–49 years old that collects information, for instance, on their health and their children’s health. The response rates tend to be high; typically exceeding 90% [21,23]. The analysis of feeding was restricted to one child per interviewed mother, where the child was 6–23 months old and living with the mother, while the analysis that focused on vitamin A supplementation only included all children 6–59 months.

### 2.2. Variables

#### Minimum Dietary Diversity and Consumption of Vitamin-A-Rich Food

The Demographic and Health Surveys record information from mothers on food given to the child 24 h before the survey. This information is only recorded for one child per household, typically the youngest, and only for children under two years old who reside with the interviewed mother. We excluded children less than six months old, which is before the weaning period starts. Several standardized food categories are used in all surveys, but the surveys often also include the consumption of country-specific foods. We grouped these foods into eight categories: (1) breastmilk; (2) grains, roots, and tubers; (3) legumes and nuts; (4) dairy products (infant formula, milk, yogurt, and cheese); (5) flesh foods (meat, fish, poultry, and liver/organ meats); (6) eggs; (7) vitamin-A-rich fruits and vegetables; and (8) other fruits and vegetables. Children who consumed from at least five of these eight food groups were considered to have achieved minimum dietary diversity. Children were considered to have consumed vitamin-A-rich foods if they consumed pumpkin, carrots, squash, sweet potatoes, mangoes, papayas, or other vitamin-A-rich fruit and vegetables; any dark green leafy vegetables; eggs; meat (including organ meat); or fish [22]. An example of the amount of vitamin A found in some of these foods is shown in Table 1.

### 2.3. Vitamin A Supplementation

WHO recommends giving children 6–11 months old a vitamin A dose of 100,000 international units (IU) and children 12–59 months a dose of 200,000 IU every 4–6 months in places where vitamin A deficiency is a public health problem [25]. The Demographic and Health Surveys record from mothers whether their children under five received a dose of vitamin A supplementation at some point over the six months preceding the survey. First, we limited the sample to firstborn children 6–23 months old living with the mother, the same sample as for child feeding, when shown with results for child feeding. We also analyzed all children 6–59 months old to compare vitamin A supplementation for children 6–23 months with children 24–59 months old.

### 2.4. Methods

We estimated the percentage of children 6–23 months old receiving the minimum dietary diversity, vitamin-A-rich foods, vitamin A supplementation, and either vitamin-A-rich foods or vitamin A supplementation. These percentages were further broken down by sex, living standards, and urban–rural residency. Living standards were measured using the household wealth index provided with the DHS: we compared the poorest and wealthiest 20% of households [26]. The relative differences according to sex, living standards, and urban–rural residency were estimated using bivariate Poisson regression models of each outcome on each group indicator. The exponent of the coefficient for the group indicator (i.e., being female, poorest, or rural) gave the relative difference, e.g., the ratio of the percentage of males receiving the minimum dietary diversity to the percentage of females receiving the minimum dietary diversity.

We further calculated the coverage of vitamin A supplementation for children 6–23 months old and 24–59 months old. We also calculated the (relative) difference in coverage between the two age groups, which was estimated using a bivariate Poisson model of an indicator for having received vitamin A supplementation on an indicator for being 24–59 months old (as opposed to 6–23 months). The exponent for the coefficient of the age indicator gave the relative difference, that is, the ratio of the percentage of children 24–59 months receiving vitamin A supplementation to the percentage of children 6–23 months receiving vitamin A supplementation. We show the results for age differences in the coverage of vitamin A supplementation by sex, living standards, and urban–rural residency in the supplement.

We calculated all results for our pooled sample, UN regions, World Bank income groups, and individual countries. Our data included 5 countries in East Asia and the Pacific, 13 in Eastern and Southern Africa, 2 in Europe and Central Asia, 4 in Latin America and the Caribbean, 3 in the Middle East and North Africa, 6 in South Asia, and 18 in West and Central Africa (see the countries in each region in tables in the supplement).

All estimates were weighted using sampling weights that were rescaled such that the sum of the weights for the complete sample of alive children 0–59 months old in each survey summed up to the under-5-year-old population at the time of the survey according to population data from UN Population division, which is available at five-year intervals (with intervening years interpolated using linear interpolation) [27]. All 95% confidence intervals (CI) were adjusted for clustering at the PSU level. We referred to relative differences as statistically significant if the 95% confidence intervals did not contain one (i.e., a 5% significance level). Note that we did not adjust the significance testing for multiple comparisons, and some estimates can be expected to be statistically significant due to sampling error.

## 3. Results

### 3.1. Feeding and Vitamin A

In the pooled sample, 22% (95% CI: 22, 23) of children 6–23 months old received the minimum dietary diversity in the 24 h before the survey, 55% (95% CI: 54, 55) received vitamin-A-rich foods in the 24 h before the survey, 59% (95% CI: 58, 59) received vitamin A supplementation within six months before the survey, and 78% (95% CI: 78, 79) received either vitamin-A-rich foods or vitamin A supplementation (Figure 1 and Appendix A). Therefore, just over one in five children missed both vitamin-A-rich foods and vitamin A supplementation.

Latin America and Caribbean and East Asia and Pacific regions had the highest percentage of children receiving the minimum dietary diversity—48% (95% CI: 47, 50) and 47% (95% CI: 45, 48), respectively, as well as the highest percentage receiving either vitamin-A-rich food or supplementation at 88% (95% CI: 87, 89) and 90% (95% CI: 89, 91), respectively. East Asia and the Pacific also had the greatest share of children receiving vitamin-A-rich foods and vitamin A supplementation separately—82% (95% CI: 80, 83) and 65% (95% CI: 63, 67), respectively.

The lowest percentage of children receiving minimum dietary diversity was observed in West and Central Africa—16% (95% CI: 15, 17), Eastern and Southern Africa—20% (95% CI: 19, 21), and South Asia—20% (95% CI: 19, 20). The lowest share of children receiving either vitamin-A-rich foods or vitamin A supplementation was in the Middle East and North Africa—69% (95% CI: 68, 71) and South Asia—76% (95% CI: 76, 77). South Asia also had the lowest share receiving vitamin-A-rich foods—46% (95% CI: 45, 47), and the Middle East and North Africa, in addition to having the lowest share receiving either vitamin-A-rich food or supplementation, also had the lowest share of children receiving vitamin A supplementation—28% (95% CI: 27, 29).

A gradient was observed by income groups: Low-income countries had the lowest percentage receiving the minimum dietary diversity—18% (95% CI: 18, 19), followed by lower-middle-income countries—24% (95% CI: 23, 24), while upper-middle-income countries had the greatest percentage—39% (95% CI: 36, 42). However, lower-middle-income countries had the lowest share of children receiving either vitamin-A-rich foods or supplementation—77% (95% CI: 77, 78), followed by low-income countries—80% (95% CI: 79, 81), while upper-middle-income countries had the greatest percentage—90% (95% CI: 88, 92).

Ethiopia and Guinea had the lowest share of children receiving the minimum dietary diversity—4% (95% CI: 4, 6) and 6% (95% CI: 5, 8), respectively (Figure 2 and Appendix A). Ethiopia and Guinea also had the lowest share of children receiving vitamin-A-rich foods—26% (95% CI: 23, 29) and 27% (95% CI: 24, 31), respectively. Further, Guinea had the lowest share of children receiving either vitamin-A-rich foods or supplementation—53% (95% CI: 49, 57), followed by Ethiopia—62% (95% CI: 59, 65). Egypt had the lowest share receiving vitamin A supplementation—21% (95% CI: 19, 22), followed by Dominican Republic—33% (95% CI: 30, 37). The greatest share of children receiving the minimum dietary diversity was observed in the Maldives—69% (95% CI: 65, 74) and in Honduras—61% (95% CI: 59, 63). The greatest share of children receiving vitamin-A-rich foods was observed in the Maldives—88% (95% CI: 85, 91), followed by Papua New Guinea—87% (95% CI: 85, 89). Rwanda and Namibia had the highest share of children receiving vitamin A supplementation—88% (95% CI: 86, 89) and 87% (95% CI: 85, 89), respectively. Burundi had the highest observed share of children receiving either vitamin-A-rich foods or supplementation—96% (95% CI: 95, 97), despite having a relatively low share of receiving dietary diversity (17%; 95% CI: 15, 19). As well as having a high share receiving the minimum dietary diversity, Honduras also has the second-highest share of children receiving either vitamin-A-rich foods or supplementation—96% (95% CI: 95, 97).

### 3.2. Differences by Sex

A lower share of males received the minimum dietary diversity than females in all regions, as well as in our pooled sample. However, the relative difference was small and only statistically significant in West and Central Africa, where the relative difference was 0.92 (95% CI: 0.86, 0.98), indicating that 8% fewer males received minimum dietary diversity (Figure 3 and Appendix A and Appendix A).

Except for the Middle East and North Africa, all regions showed a lower share of males receiving vitamin-A-rich foods than females. However, the sex differences were minor and not statistically significant in any region. The sex difference was small but marginally statistically significant in the pooled sample, where the relative difference was 0.99 (95% CI: 0.97, 1.00), indicating that 1% fewer males received vitamin-A-rich foods. Sex differences in the share of children receiving vitamin A supplementation and children receiving either vitamin-A-rich foods or supplementation were also small and not statistically significant in any region or the pooled sample.

One country had a statistically significant sex difference for the minimum dietary diversity, namely, Yemen, where 15% fewer males received the minimum dietary diversity (0.85; 95% CI: 0.72, 0.99; Figure 4 and Appendix A and Appendix A). Three countries had a statistically significant sex difference regarding the consumption of vitamin-A-rich foods, with females consuming more than males: India (0.97; 95% CI: 0.95, 1.00), Comoros (0.89; 95% CI: 0.80, 0.99), and South Africa (0.88; 95% CI: 0.80, 0.96). Three countries had a statistically significant greater share of males receiving vitamin A supplementation: Haiti (1.11; 95% CI: 1.00, 1.23), Nepal (1.09; 95% CI: 1.02, 1.17), and Uganda (1.08; 95% CI: 1.01, 1.17). For three countries, the share receiving vitamin A supplementation was statistically significantly lower for males: Congo DR (0.94; 95% CI: 0.89, 0.99), Cambodia (0.92; 95% CI: 0.86, 0.98), and Bangladesh (0.91; 95% CI: 0.84, 0.99). Three countries had a statistically significant sex difference regarding receiving either vitamin-A-rich foods or supplementation: favoring males in Nepal (1.06; 95% CI: 1.01, 1.12) and females in Niger (0.95; 95% CI: 0.90, 0.99) and Indonesia (0.97; 95% CI: 0.95, 1.00).

### 3.3. Differences by Living Standards

In the pooled sample, 15% (95% CI: 15, 16) of children in the 20% of households with the worst living standards (poorest) received the minimum dietary diversity, while that share was 31% (95% CI: 30, 33) in the 20% of households with the best living standards (wealthiest), leading to a relative difference of 0.49 (95% CI: 0.47, 0.52; Figure 5 and Appendix A and Appendix A). The difference by living standards was statistically significant in all regions, except for Europe and Central Asia, where the share of children in the poorest households receiving the minimum dietary diversity was still 12% (95% CI: 0.71, 1.10) lower than in the wealthiest households. The difference was greatest in West and Central Africa, where the relative difference was 0.33 (95% CI: 0.28, 0.38), indicating that 67% fewer children in the poorest households received the minimum dietary diversity than in the wealthiest households. A gradient was observed by country income groups, where the relative disadvantage of the poorest was the greatest for the low-income countries—0.39 (95% CI: 0.36, 0.43), followed by lower-middle-income countries—0.52 (95% CI: 0.49, 0.56), while being the smallest in upper-middle-income countries—0.66 (95% CI: 0.51, 0.86).

All regions and the pooled sample showed a lower share of children in the poorest households receiving vitamin-A-rich foods. However, the difference was not statistically significant in Europe and Central Asia (95% CI: 0.75, 1.02). Similarly, the poorest children also had a lower share receiving vitamin A supplementation in all regions and overall, although the difference was not statistically significant in the Middle East and North Africa (95% CI: 0.75, 1.04) and Europe and Central Asia (95% CI: 0.81, 1.12).

In the pooled sample, 72% (95% CI: 71, 73) of the poorest children received either vitamin-A-rich foods or supplementation, which was 14% (95% CI: 0.85, 0.87) lower than in the wealthiest households, where that share was 84% (95% CI: 83, 85). The difference by living standards was greatest in West and Central Africa, where 68% (95% CI: 66, 70) of children in the poorest households received either vitamin-A-rich foods or vitamin A supplementation, which was 0.77 (95% CI: 0.74, 0.80) times that of the wealthiest, or 23% lower. The relative difference was statistically significant in all regions except Europe and Central Asia, where the relative difference was only 0.98 (95% CI: 0.89, 1.09).

Three countries had a greater share of children in the poorest household receiving minimum dietary diversity than in the wealthiest households, although these differences were not statistically significant (Figure 6 and Appendix A and Appendix A). Forty-four countries had statistically significantly lower minimum dietary diversity in the poorest households. The largest relative difference was observed in Ethiopia (0.12; 95% CI: 0.05, 0.29), followed by Cote d’Ivoire (0.15; 95% CI: 0.07, 0.33) and Niger (0.17; 95% CI: 0.01, 0.28).

In nine countries, children in the poorest households were more likely to receive vitamin-A-rich foods, although the relative difference was only statistically significant in two countries: Congo (1.13; 95% CI: 1.02, 1.26) and Mozambique (1.09; 95% CI: 1.01, 1.19). The poorest were statistically significantly less likely to receive vitamin-A-rich foods in 29 countries with the greatest relative difference observed in Guinea (0.35; 95% CI: 0.25, 0.48), Niger (0.46; 95% CI: 0.36, 0.56), and Ethiopia (0.46; 95% CI: 0.35, 0.60).

In eight countries, children in the poorest households were more likely to receive vitamin A supplementation, although the relative difference was only statistically significant in one country: Cameroon (1.14; 95% CI: 1.04, 1.26). In 23 countries, the relative difference indicated a statistically significant lower coverage of vitamin A supplementation for the poorest children compared to the wealthiest, with the greatest relative difference observed in Nigeria (0.29; 95% CI: 0.25, 0.35) and Papua New Guinea (0.42; 95% CI: 0.31, 0.57).

Six countries had a greater share of children in the poorest households receiving either vitamin-A-rich foods or supplementation than in the wealthiest households, although the differences were not statistically significant. The relative difference by living standards in receiving either vitamin-A-rich foods or supplementation was statistically significant in 29 countries. The greatest relative difference was observed in Guinea (0.53; 95% CI: 0.44, 0.65), followed by Nigeria (0.61; 95% CI: 0.57, 0.66).

### 3.4. Urban–Rural Differences

In the pooled sample, 19% (95% CI: 18, 19) of children in rural areas received the minimum dietary diversity, while the share was 30% (95% CI: 30, 31) in urban areas, a relative difference of 0.62 (95% CI: 0.60, 0.65; Figure 7 and Appendix A and Appendix A). All regions except Europe and Central Asia (95% CI: 0.88, 1.19) had a statistically significantly lower share of children receiving the minimum dietary diversity in rural areas than urban areas. The largest relative difference was observed in Eastern and Southern Africa, where 16% (95% CI: 15, 17) of children in rural areas received the minimum dietary diversity while 32% (95% CI: 30, 34) received it in urban areas, or a relative difference of 0.49 (95% CI: 0.44, 0.53).

In the pooled sample and all regions except Europe and Central Asia, children in rural areas were less likely to receive vitamin-A-rich foods. The pooled sample and all regions except Europe and Central Asia and the Middle East and North Africa showed lower coverage of vitamin A supplementation in rural areas compared to urban areas.

About three-quarters (76%; 95% CI: 76, 77) of children in rural areas in the pooled sample received either vitamin-A-rich foods or vitamin A supplementation, while 83% (95% CI: 82, 83) did so in urban areas, for a relative difference of 0.92 (95% CI: 0.92, 0.94). Except for two countries in Europe and Central Asia, all regions showed a lower share of vitamin A consumption in rural areas than urban areas. The relative difference was greatest in West and Central Africa—0.88 (95% CI: 0.85, 0.90), where 74% (95% CI: 73, 75) of children in rural areas and 84% (95% CI: 83, 86) of children in urban areas received either vitamin-A-rich foods or vitamin A supplementation.

The relative urban–rural difference in minimum dietary diversity was not statistically significant in ten countries, four of which indicated greater minimum dietary diversity in rural areas (Figure 8 and Appendix A and Appendix A). The relative difference in the minimum dietary diversity was the greatest in Niger (0.21; 95% CI: 0.16, 0.27), followed by Guinea (0.28; 95% CI: 0.17, 0.47). Ten countries had a greater share of children in rural areas receiving vitamin-A-rich foods than in urban areas; this difference was statistically significant in three countries: Congo (1.13; 95% CI: 1.05, 1.21), Cambodia (1.10; 95% CI: 1.02, 1.19), and Burundi (1.06; 95% CI: 1.00, 1.12). The share receiving vitamin-A-rich foods was statistically significantly lower in rural areas in 26 countries, with the greatest relative differences observed in Niger (0.55; 95% CI: 0.49, 0.62) and Guinea (0.57; 95% CI: 0.45, 0.72).

Rural households had a greater share that received vitamin A supplementation in 20 countries, where the relative difference was statistically significant in two countries: South Africa (1.10; 95% CI: 1.00, 1.20) and Cameroon (1.09; 95% CI: 1.01, 1.17). Twenty-two countries had a statistically significantly lower share that received vitamin A supplementation in rural households, with the greatest relative difference observed in Nigeria (0.62; 95% CI: 0.55, 0.70) and Guinea (0.65; 95% CI: 0.54, 0.79).

Nine countries had a greater share of children in rural areas that received either vitamin-A-rich foods or supplementation than in urban areas, although these differences were small and not statistically significant, except for in Congo (1.05; 95% CI: 1.00, 1.09). Twenty-four countries had a statistically significantly lower share of children that received either vitamin-A-rich foods or supplementation in rural areas compared with urban areas. The relative difference was the greatest in Guinea (0.72; 95% CI: 0.62, 0.83), followed by Nigeria (0.81; 95% CI: 0.77, 0.86) and Niger (0.81; 95% CI: 0.77, 0.86).

### 3.5. Comparison of Vitamin A Supplementation before and after Age Two Years

In the pooled sample, 59% of both children 6–23 months old (95% CI: 58, 60) and 24–59 months old (95% CI: 59, 60) had received vitamin A supplementation in the six months before the survey (Figure 9 and Appendix A). In East Asia and the Pacific, 65% (95% CI: 63, 67) of children under two years old received vitamin A supplementation while 70% (95% CI: 69, 71) of those two years and older did so, leading to a statistically significant 8% (95% CI: 1.05, 1.11) difference. In Europe and Central Asia, 55% (95% CI: 52, 58) of children under two years old received a vitamin A dose while 59% (95% CI: 56, 61) of children two years and older received a dose, leading to a statistically significant difference of 6% (95% CI: 1.01, 1.11). In Eastern and Southern Africa, 63% of children under age two years (95% CI: 62, 64) and over age two years (95% CI: 62, 64) received vitamin A supplementation. In South Asia, 61% of children under two years old (95% CI: 61, 62) and over two (95% CI: 61, 62) received vitamin A supplementation. In West and Central Africa, 55% (95% CI: 54, 57) of children under two years old received vitamin A supplementation, while 56% (95% CI: 54, 57) of children over two years old received it, leading to a non-statistically significant relative difference of 1.01 (95% CI: 0.99, 1.02).

In two regions, children under two years old were more likely to have received vitamin A supplementation, a result that was statistically significant in both: in the Middle East and North Africa, 28% (95% CI: 27, 29) of children under two years old received vitamin A supplementation, while 25% (95% CI: 23, 26) of those over two years old did so, leading to a relative difference of 0.88 (95% CI: 0.84, 0.93); in Latin America and the Caribbean, 58% (95% CI: 56, 60) of children under two years old received vitamin A supplementation, while 43% (95% CI: 23, 26) of those over two years old did, for a relative difference of 0.74 (95% CI: 0.72, 0.77).

Nineteen countries had a statistically significant higher share of children 24–59 months old that received vitamin A supplementation than for children 6–23 months, ranging from 31% (95% CI: 1.19, 1.45) greater in the Maldives and 23% (95% CI: 1.17, 1.29) greater in Bangladesh, to 3% greater in Togo (95% CI: 1.01, 1.05) and Sierra Leone (95% CI: 1.01, 1.06; Figure 10 and Appendix A). Fifteen countries had a statistically significantly lower share of vitamin A supplementation for children 24–59 months compared with 6–23 months, ranging from 64% (95% CI: 0.29, 0.44) lower in Jordan and 36% (95% CI: 0.58, 0.69) lower in Papua New Guinea, to 5% (95% CI: 0.94, 0.96) lower in India and 6% lower in Tanzania (95% CI: 0.88, 1.00), Burkina Faso (95% CI: 0.91, 0.96), and Kenya (95% CI: 0.91, 0.96).

## 4. Discussion

This analysis showed four main findings regarding feeding and vitamin A supplementation in 51 low- and middle-income countries. First, 78% of children aged 6 months to two years in the pooled sample received either vitamin-A-rich foods or vitamin A supplementation ranging from 53% to 96% across countries. The 22% of children that received neither were the children at highest risk of suffering negative health outcomes as a result of vitamin A deficiency; the analysis presented here showed that children residing in the poorest households and rural areas were at more risk, and there was also variation between countries ranging from 4–47% of children receiving neither vitamin-A-rich foods nor supplementation. Second, only 22% of children in our sample received the minimum dietary diversity during the weaning period of 6–23 months old; however, the percentage ranged widely between 4% and 69%, across countries. Third, there were no clear sex differences in the minimum dietary diversity, consumption of vitamin-A-rich foods, or vitamin A supplementation. In contrast, differences according to living standards and urban–rural residency were common and large, especially for the minimum dietary diversity: Overall, children in the 20% of households with the lowest living standards had a 51% lower probability of receiving the minimum dietary diversity and 14% lower likelihood of receiving either vitamin-A-rich foods or supplementation than children in the 20% of households with the greatest living standards. Children in rural households had a 48% lower probability of receiving the minimum dietary diversity and an 8% lower share of children getting either vitamin-A-rich foods or supplementation than children in urban households. 

The fourth main finding was that, overall, there were no differences in the coverage of vitamin A supplementation between younger (6–23 months old) and older (24–59 months old) children. A plausible explanation was that mass campaigns, which have been the main delivery mode for vitamin A supplementation, reached younger and older children to the same extent. However, as countries transition away from delivery in campaigns toward routine health system contacts, it may be more difficult to reach older children. Moreover, the pooled estimates obscured significant variations: for West and Central Africa, Eastern and Southern Africa, and South Asia, we observed no difference in vitamin A supplementation between children under and over two years old, while East Asia and the Pacific and Europe and Central Asia had 7% and 6% greater coverage among children over two years old, respectively, and Latin America and the Caribbean and the Middle East and North Africa had, respectively, 26% and 12% lower coverage among children over two years. Further, 30 countries had higher coverage in children over two years old (statistically significant in 21 countries), while 21 countries had lower coverage in older children (statistically significant in 15 countries).

With more than a fifth of children aged 6–23 months receiving neither vitamin-A-rich foods nor vitamin A supplementation, there continues to be a need for nutrition interventions that effectively address vitamin A deficiency. However, the feasibility of prioritizing interventions in high-risk children depends on numerous factors, including the cost of interventions, marginal costs of delivery, the clustering of poor health outcomes geographically, and the accuracy of the identification of risk. A study from 1997 suggested that targeting vitamin A supplementations toward children at risk based on malnutrition measures was less effective and efficient than universal distribution [28]. However, age is another more easily identifiable risk factor: adverse child health outcomes, such as mortality and wasting, are much more prevalent among younger children, and adverse exposures are particularly detrimental before the age of two years. The targeting of vitamin A supplementation based on age has recently been suggested as an efficient way to improve the coverage of the most vulnerable children: young children are easy to identify and should have routine interactions with healthcare systems (e.g., for immunization, growth and health monitoring, and parental counseling), which can be used to also administer vitamin A supplementations [29]. However, it should be noted that some researchers question the true efficacy and safety of administering universal high-dose vitamin A supplementation for any age group [30]. For example, vitamin A may be an antagonist of vitamin D, leading to bone demineralization [31]. Studies have also provided tentative evidence of a greater effect of vitamin A supplementation on growth for children older than two years than children younger than two years, possibly due to extended breastfeeding protecting younger children from vitamin A deficiencies [32].

### Limitations

The use of observational data, which may include inaccuracies, was a limitation of this study. Information on feeding, vitamin A supplementation, and children’s ages was reported by the mother from memory and could therefore suffer from recall bias. Survey participants may further confuse vitamin A supplementation with other interventions, primarily the polio vaccine, which was similarly administered as drops. Further, we did not study the quantity and quality of foods in each food category used to construct the measures for minimum dietary diversity and consumption of vitamin-A-rich foods. Our estimates did not capture the seasonality of foods, particularly vitamin-A-rich foods and vegetables [33,34]. Finally, we had incomplete coverage of countries and an imbalance in the coverage of regions, with few countries representing the Middle East and North Africa, Latin America and the Caribbean, and, especially, Europe and Central Asia, which only included two countries in this study.

## 5. Conclusions

These results highlighted the longer-term need to improve dietary diversity in most of the countries in this study. In the short-to-medium term, there is an urgent need to prioritize children that have an inadequately diverse diet for vitamin A supplementation. The coverage of vitamin A supplementation for children 6–59 months old (especially the youngest children) every 4–6 months thus needs to be improved in most countries, particularly in countries where the consumption of vitamin-A-rich foods is low and in the poorest households and rural areas. This analysis found no current discernable difference in reach between children over and under two years old by vitamin A supplementation programs. However, this might change as countries move to more sustainable delivery in established routine health system contacts. Unfortunately, there is no information on diet diversity or the consumption of vitamin-A-rich foods in children over two years old. However, children under two years old need particular attention and to be ensured complete coverage to boost immune function due to their vulnerability and long-term consequences of infections during this period. Ensuring adequate dietary intake and coverage of vitamin A supplementation for the most vulnerable children, namely, those under two years old, can bring significant gains in child health in low- and middle-income countries.

## Figures and Tables

**Figure 1 nutrients-14-00188-f001:**
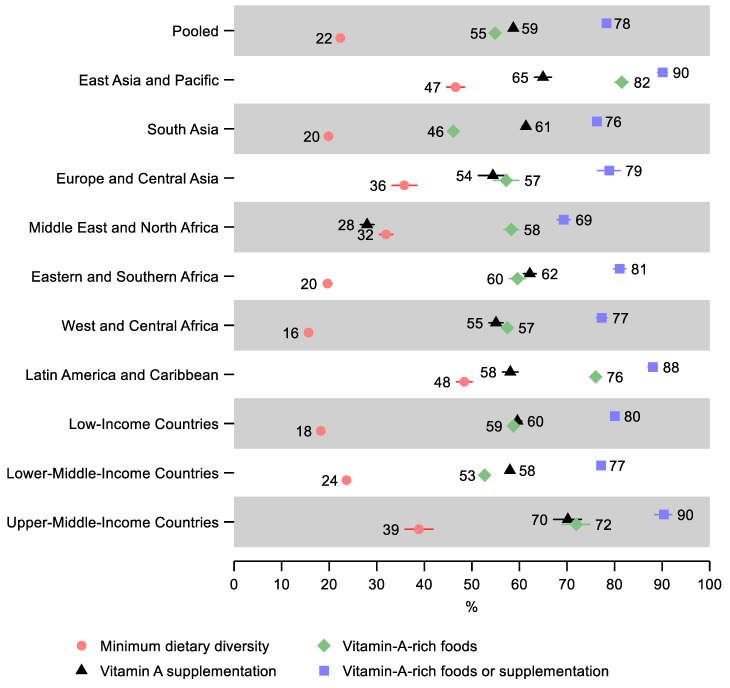
Percentage of children 6–23 months old that received the minimum dietary diversity, vitamin-A-rich foods, vitamin A supplementation, and either vitamin-A-rich foods or vitamin A supplementation in low- and middle-income countries. Notes: 95% confidence intervals are shown. Some confidence intervals may be narrower than the symbol for the estimate and therefore not visible. See Appendix A for the tabulated estimates and confidence intervals.

**Figure 2 nutrients-14-00188-f002:**
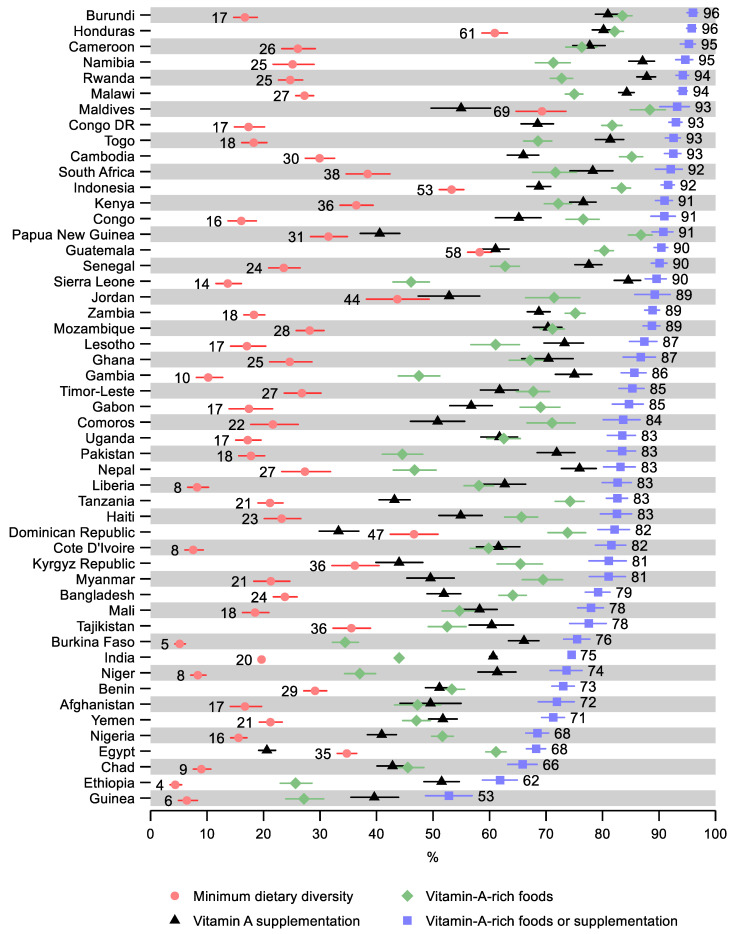
Percentage of children 6–23 months old that received the minimum dietary diversity, vitamin-A-rich foods, vitamin A supplementation, and either vitamin-A-rich foods or vitamin A supplementation by country. Notes: 95% confidence intervals are shown. Countries are ordered from largest to smallest percentage of children receiving either vitamin-A-rich foods or supplementation. Some confidence intervals may be narrower than the symbol for the estimate and therefore not visible. See Appendix A for the tabulated estimates and confidence intervals.

**Figure 3 nutrients-14-00188-f003:**
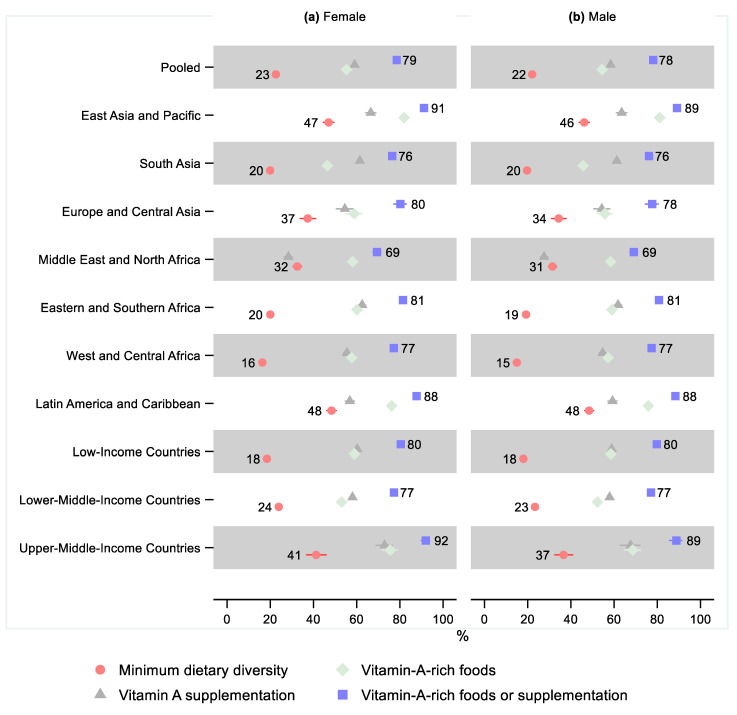
Percentage of children 6–23 months old that received the minimum dietary diversity, vitamin-A-rich foods, vitamin A supplementation, and either vitamin-A-rich foods or vitamin A supplementation in low- and middle-income countries by sex. Notes: 95% confidence intervals are shown. Some confidence intervals may be narrower than the symbol for the estimate and therefore not visible. See Appendix A for the tabulated estimates and confidence intervals. See Appendix A for the plotted relative differences.

**Figure 4 nutrients-14-00188-f004:**
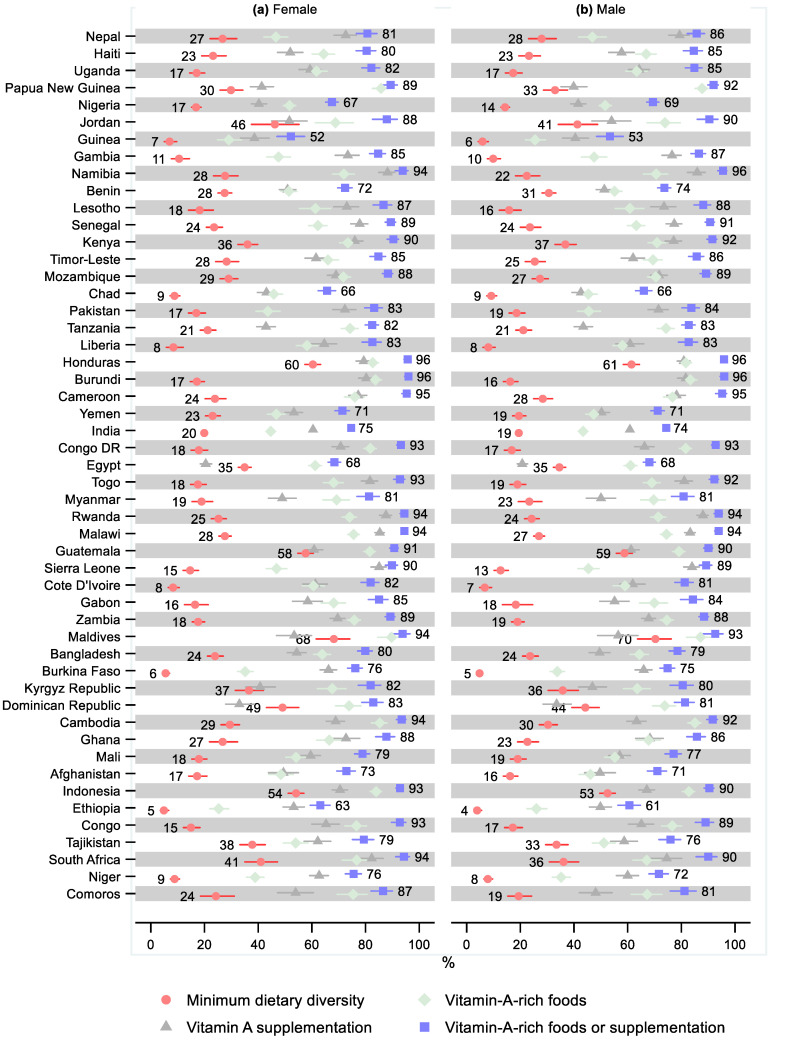
Percentage of children 6–23 months old that received the minimum dietary diversity, vitamin-A-rich foods, vitamin A supplementation, and either vitamin-A-rich foods or vitamin A supplementation by sex and country. Notes: 95% confidence intervals are shown. Countries are ordered from largest to smallest relative sex difference in the percentage of children that received either vitamin-A-rich foods or supplementation. Some confidence intervals may be narrower than the symbol for the estimate and therefore not visible. See Appendix A for the tabulated estimates and confidence intervals. See Appendix A for the plotted relative differences.

**Figure 5 nutrients-14-00188-f005:**
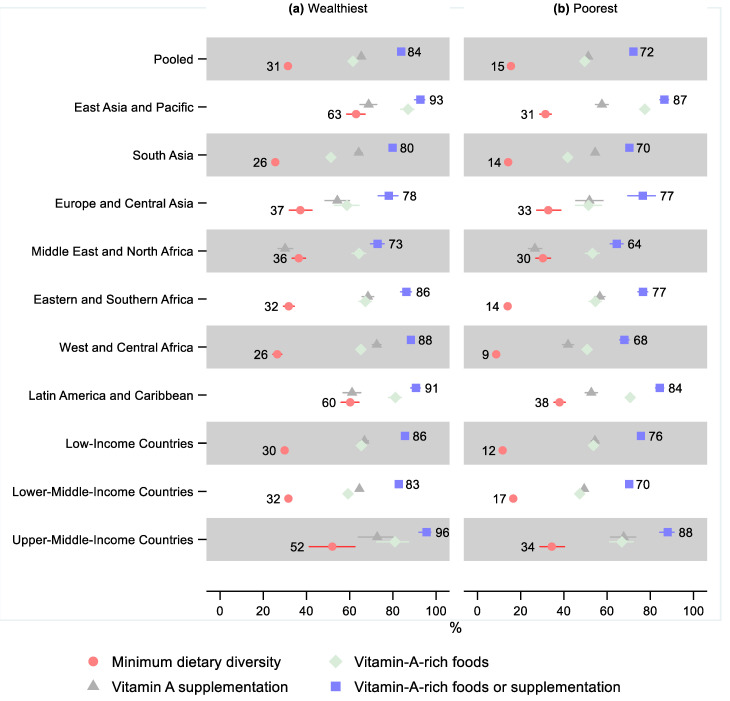
Percentage of children 6–23 months old that received the minimum dietary diversity, vitamin-A-rich foods, vitamin A supplementation, and either vitamin-A-rich foods or vitamin A supplementation in low- and middle-income countries by living standards. Notes: 95% confidence intervals are shown. Some confidence intervals may be narrower than the symbol for the estimate and therefore not visible. See Appendix A for the tabulated estimates and confidence intervals. See Appendix A for the plotted relative differences.

**Figure 6 nutrients-14-00188-f006:**
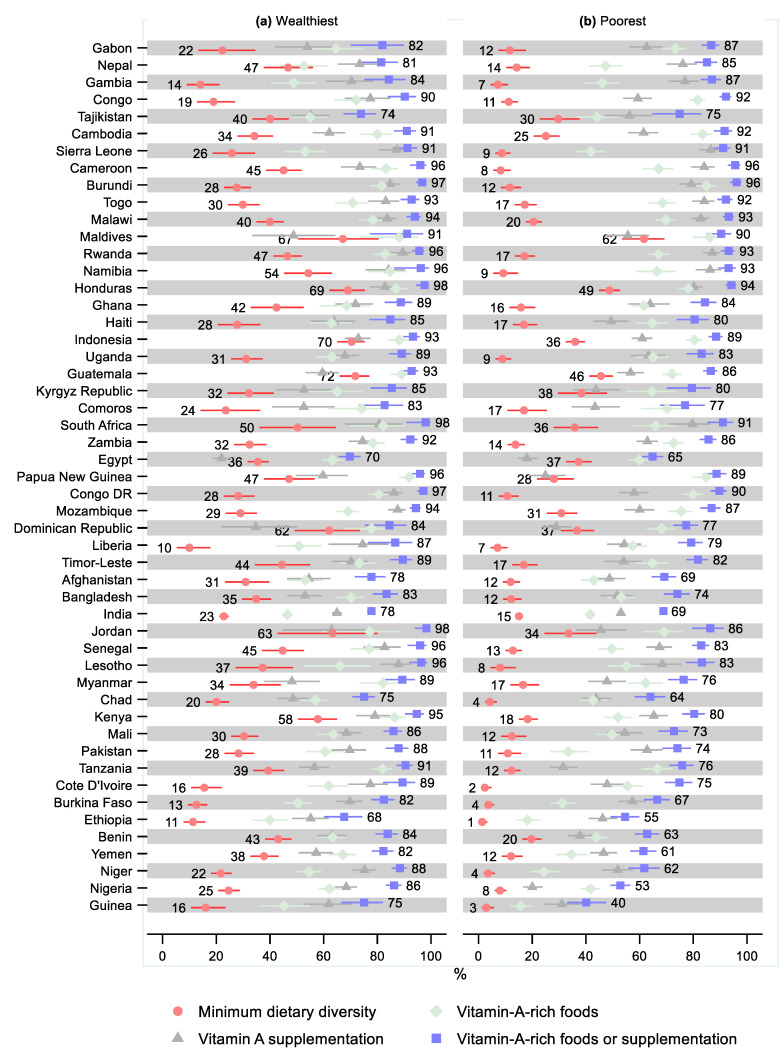
Percentage of children 6–23 months old that received minimum dietary diversity, vitamin-A-rich foods, vitamin A supplementation, and either vitamin-A-rich foods or vitamin A supplementation by living standards and country. Notes: 95% confidence intervals are shown. Countries are ordered from largest to smallest relative living standards difference in percentage of children that received either vitamin-A-rich foods or supplementation. Some confidence intervals may be narrower than the symbol for the estimate and therefore not visible. See Appendix A for the tabulated estimates and confidence intervals. See Appendix A for the plotted relative differences.

**Figure 7 nutrients-14-00188-f007:**
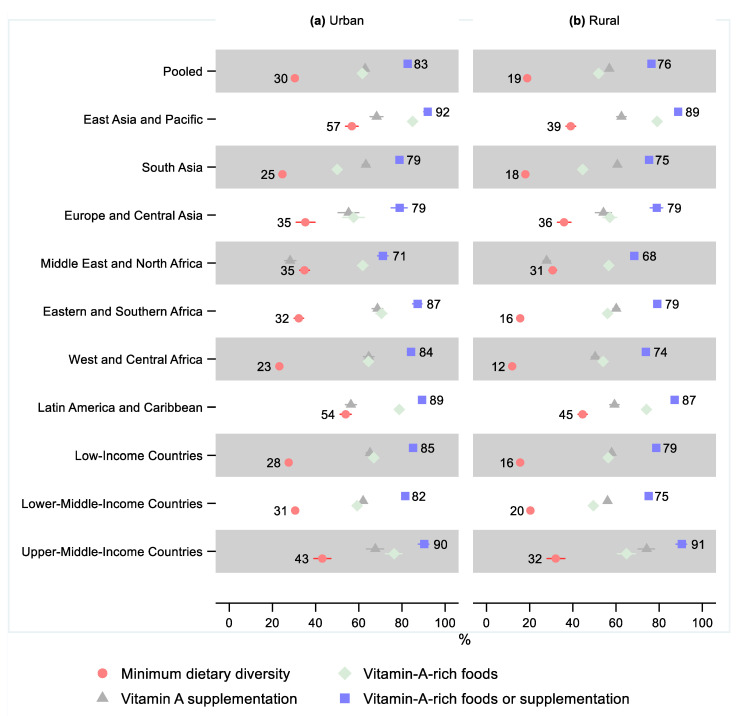
Percentage of children 6–23 months old that received the minimum dietary diversity, vitamin-A-rich foods, vitamin A supplementation, and either vitamin-A-rich foods or vitamin A supplementation in low- and middle-income countries by urban–rural residency. Notes: 95% confidence intervals are shown. Some confidence intervals may be narrower than the symbol for the estimate and therefore not visible. See Appendix A for the tabulated estimates and confidence intervals. See Appendix A for the plotted relative differences.

**Figure 8 nutrients-14-00188-f008:**
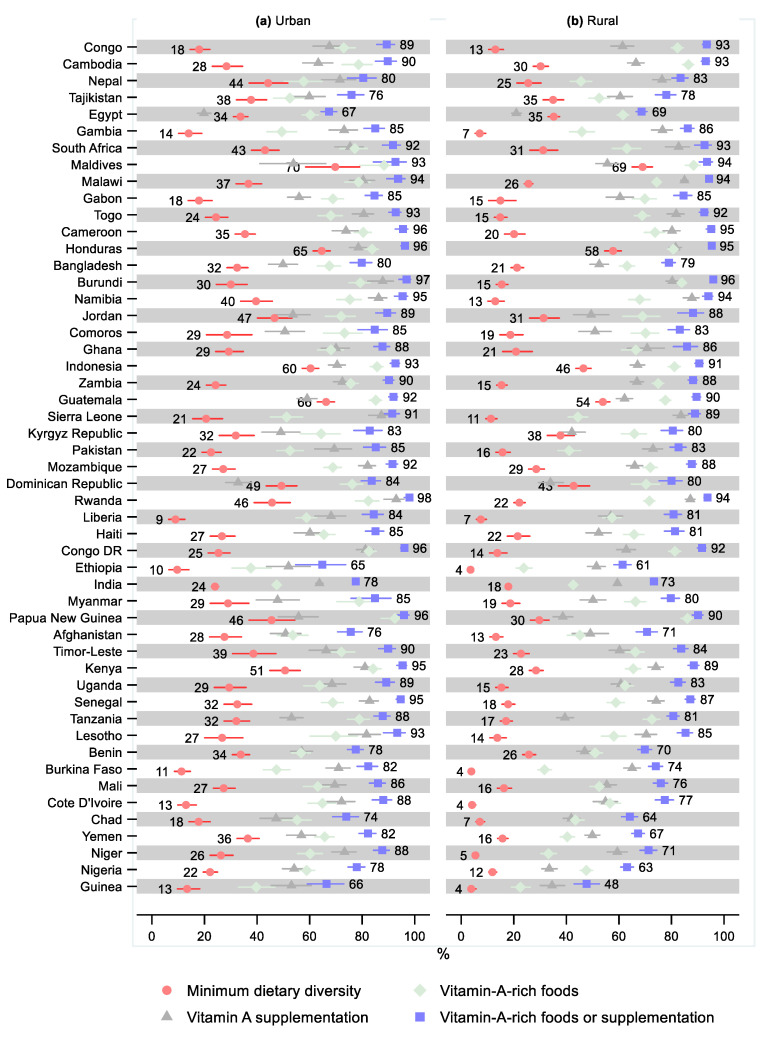
Percentage of children 6–23 months old that received minimum dietary diversity, vitamin-A-rich foods, vitamin A supplementation, and either vitamin-A-rich foods or vitamin A supplementation by urban–rural residency and country. Notes: 95% confidence intervals are shown. Countries are ordered from largest to smallest relative urban–rural residency difference in percentage of children that received either vitamin-A-rich foods or supplementation. Some confidence intervals may be narrower than the symbol for the estimate and therefore not visible. See Appendix A for the tabulated estimates and confidence intervals. See Appendix A for the plotted relative differences.

**Figure 9 nutrients-14-00188-f009:**
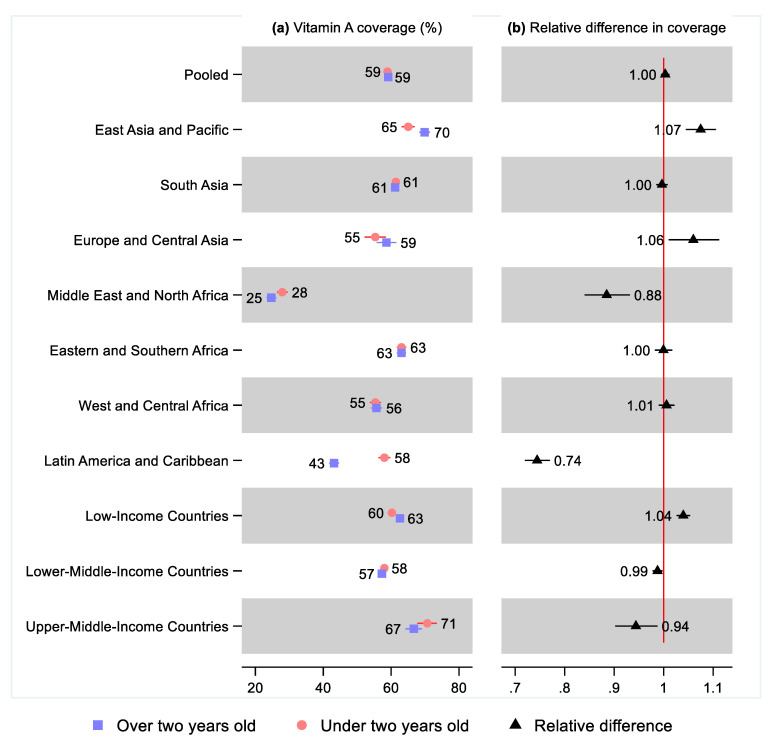
Percentage of children under two years old (6–23 months old) and over two years old (24–59 months old) that received vitamin A supplementation in low- and middle-income countries. Notes: 95% confidence intervals are shown. Some confidence intervals may be narrower than the symbol for the estimate and are therefore not visible. See Appendix A for the tabulated estimates and confidence intervals. See Appendix A and Appendix A for the differences by sex, living standards, and urban–rural residency.

**Figure 10 nutrients-14-00188-f010:**
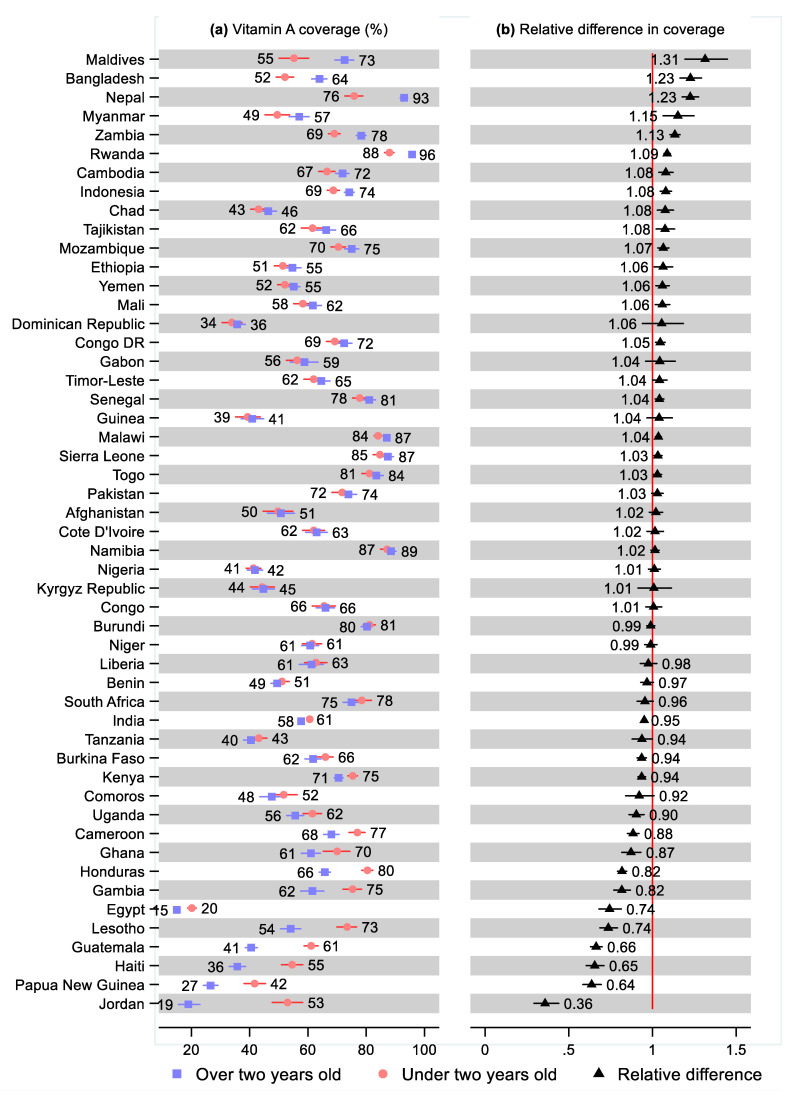
Percentage of children under two years old (6–23 months) and over two years old (24–59 months) that received vitamin A supplementation by country. Notes: 95% confidence intervals are shown. Countries are ordered from largest to smallest relative age difference. Some confidence intervals may be narrower than the symbol for the estimate and are therefore not visible. See Appendix A for the tabulated estimates and confidence intervals. See Appendix A and Appendix A for the differences by sex, living standards, and urban–rural residency.

**Table 1 nutrients-14-00188-t001:** Examples of vitamin-A-rich foods.

Foot Item	Vitamin A (IU) per 100 g
Pumpkin	8513
Carrots	16,706
Squash (e.g., butternut)	10,630
Sweet potatoes	14,187
Mangoes	1082
Papayas	950
Other vitamin-A-rich fruit and vegetables (e.g., red grapefruit)	1150
Dark green leafy vegetables (e.g., collards)	5019
Eggs	520
Meat (e.g., poultry)	245
Fish (e.g., herring)	93

Source: USDA, 2019 [24].

## Data Availability

DHS data are available at https://dhsprogram.com (accessed on 7 June 2021) (requiring a simple application).

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
