# Peer review of "Consumption of Vitamin-A-Rich Foods and Vitamin A Supplementation for Children under Two Years Old in 51 Low- and Middle-Income Countries"

_nutrients, 2021, doi:10.3390/nu14010188_

Round 1
Reviewer 1 Report
Comments to the Author
The paper by Karlsson et al.: Consumption of vitamin A rich foods and vitamin A supplementation for children under two in 51 low- and middle-income countries, it’s interesting enough but it needs minor revision.
- page 1, line 43: at the end of the paragraph the authors should add the reference. 'Then children should receive 41 safe and nutritionally adequate food—in terms of protein and energy as well as vitamins 42 and minerals—together with continued breastfeeding until age 23 months or longer'
- in Introduction (before Data and Methods) the Authors are strongly advised to include a Table with foods containing vitamin a specifying the amount for each
- all the figures (including those in supplementary files) should be redone because they are neither scientific nor clear
- In discussion the Authors should add a paragraph on the interaction with at least 2 other vitamins:D and K
Reviewer 2 Report
Really fantastic large dataset analysis!!! Policy makers can read all details in those supplemental tables.
I would also be interested in vitamin A rich intake or vitamin A supplementation in young children in high-income or middle-income countries. Will you also analyse those data?
Please put in all titles of the graphs that only low-income and middle-income countries are in the analysis. Otherwise, people can misinterpret those graphs, as for example where Europe is written, it is very limited information. The data of those 51 countries is not representative of all countries worldwide, so this selection should be in the legend of the graphs and figures.
Line 5 -please remove the titles as all authors have educational level. Does the last author have a first name?
Line 5 -not sure why first author also has Lund university as reference as he graduated in 2018 and is now an employee at Harvard university. Same comment for second author, as a person does work at Korea university or that person work at Harvard Center USA. I only want to warn that some persons give multiple affiliations to score more publications for various academic scoring systems.
Line 19 – add that vitamin A supplementation is important intervention for young children in low- and middle-income countries. In high-income countries, young children do get follow-on formula or growing-up milk which is already fortified with vitamin A.
Line 20 – a word is missing. Add ‘get’ or ‘consume’ or ‘have’
Line 21 – the word ‘coverage’ is a bit strange. Do you mean ‘availability’ or ‘dietary intake’?
Line 31-32 -I have difficulties with the age range 6-23 months old, as they should get (exclusively) breastfeeding according to the WHO guidelines (especially in low-income countries where clean safe water is not largely available for making infant formula). Most countries recommend at least 6 and preferably 12 months breastfeeding and then 12-23 months partially breastfeeding. So, it is the question if the lactating mothers consume vitamin A rich foods. Later in line 41-43 this weaning period with breastfeeding is explained, but I like to see this also in the abstract as breastfeeding is always the best.
Introduction part is of high level and well written and well explained.
Line 86. How much countries do have demographic and health surveys, so how much countries were excluded as they did not do the survey after 2010? It is impressive to have data from 51 countries, but please explain which countries were excluded and why.
Lines 83-99. I am curious how the translation and analysis was done as likely the national language was used in those 51 countries. Other option is that those surveys use the same coding system with the same food categories in all countries, but was this the case?
Line 120 – how is the dose of the vitamin A supplementation for young children? Is that the same dose for all those 51 countries? What is the (estimated) range?
Line 120 – “supplantation”?
Line 148 – two in Europe? I cannot believe that only two surveys are done after 2010, so what was exactly the reasons to exclude all those other countries? And which countries are in Europe in your analysis? I cannot find them.
Line 166-167 – Is this the main outcome that globally 1 in 5 young children do not consume enough vitamin A rich food or vitamin A supplementation? Then this should be in the abstract. In the abstract is now written “22% of children received minimal dietary diversity”; that is not the same wording as 1 in 5 young children do not consume enough vitamin A rich food or get a vitamin A supplementation. In lines 426-432 it is well described further, but abstract is now less clearly readable.
